# Clarification of Mining Process Water Using Electrocoagulation

Oscar Joaquín Solis-Marcial [1,*], Alfonso Talavera-López [2], José Pablo Ruelas-Leyva [3],
José Alfredo Hernández-Maldonado [4], Alfonso Najera-Bastida [1], Roberto Zarate-Gutierrez [5]
and Benito Serrano Rosales [6]

[1] Unidad Profesional Interdisciplinaria de Ingeniería Campus Zacatecas, Instituto Politécnico Nacional,
Calle Circuito del Gato 202, Zacatecas 98160, Mexico; anajerab@ipn.mx
[2] Unidad de Ciencias Químicas, Universidad Autónoma de Zacatecas, Carretera Zacatecas–Guadalajara Km. 6,
Zacatecas 98160, Mexico; talaram20@hotmail.com
[3] Facultad de Ciencias Químico Biológicas, Universidad Autónoma de Sinaloa, Culiacán 80030, Mexico;
joseruelas@uas.edu.mx
[4] Unidad Profesional Interdisciplinaria de Ingeniería Campus Guananjuato, Instituto Politécnico Nacional,
Guanajuato 36275, Mexico; jahernandezma@ipn.mx
[5] Centro de Estudios Científicos y Tecnológicos No-18, Instituto Politécnico Nacional,
Calle Circuito del Gato 202, Zacatecas 98160, Mexico; rzarate@ipn.mx
[6] Independent Researcher, Guadalupe, Zacatecas 98613, Mexico; beniser@prodigy.net.mx
* Correspondence: ojsolis@ipn.mx; Tel.: +51-4921910154

**Abstract:** A lack of fresh water is one of the most significant problems currently affecting humanity. Water scarcity also affects industries, with the mining industry being one of the most affected. One possible solution to water scarcity is the recirculation of water. Water in mining is usually treated with physicochemical methods, but in metallurgical processes, reagents are added, accumulate until reaching the point of saturation, and are often not successfully removed. In this sense, electrocoagulation has shown great efficiency in the treatment of organic contaminants, heavy metals, and metallic ions, and was applied in this study to eliminate ions and undesirable organic compounds present in mining–metallurgical process water. Furthermore, this process has shown great efficacy in relation to toxic metals like arsenic because their presence reduces the efficiency of other processes such as flotation. In this study, two types of electrodes were used: stainless steel and aluminum. The best results were achieved with stainless steel electrodes, which were able to eliminate 90% of copper ions in water. The turbidity of the water during the process was measured to determine the amount of solid present in the water, and a reduction of around 95% was observed when using aluminum electrodes. The sedimentation of clots occurred in two stages: Firstly, the coagulant was formed to trap organic matter as its size increased, until a particle size that was sufficient for settling was achieved. A zero-order kinetic model was fit for this stage of the process. Secondly, the formed clots continued to settle, and a second-order kinetic model was fit for this stage. Flotation tests were carried out on the process and electrotreated water to evaluate the recovery of Zn, Pb, Ag, and Au. An increase of 1.5% was found for gold, and an increase of 2% was found for silver, while a significant improvement was identified for zinc, augmenting recuperation by 30% when electrotreated water was used. For lead, no considerable change in recovery was observed in either form of water. The formed clots were analyzed using Scanning Electronic Microscopy, and we found that metal ions were trapped in the clots. This study demonstrates the potential of electrocoagulation for clarifying mine water, which is ordinarily very difficult to clarify.

**Keywords:** electrocoagulation; recovery water; chemical reagents; electricity

## 1. Introduction

One of the biggest problems of the 21st century is satisfying the world population's need for purified water, which is increasing day by day. Although two-thirds of the planet is water, only 2.53% is fresh water [1]. In this sense, the supply of wastewater plays an essential

role, as it could potentially be recirculated indefinitely [2]. For this reason, it is important to develop efficient and innovative techniques that compete technically, economically, and environmentally with traditional technology [3], such as biological and physicochemical processes. The most common physicochemical processes are filtration, ionic exchange, chemical precipitation, chemical oxidation, adsorption, ultrafiltration, reverse osmosis, and electrodialysis [4,5]. Several emerging modern techniques are based on electrochemistry and show competitive advantages over traditional techniques, such as electrocoagulation, electroflotation, and electrodecantation. Recently, electrocoagulation was proposed as an effective method to treat water contaminated with heavy metals, fluorine, suspended solids, restaurant residues (oils and fats), and residues from the cigarette industry [6]. Most research on electrocoagulation has shown a significant reduction in pollutants, and interest in this technology has increased on an industrial scale [7]. Electrochemical technology offers an elegant contribution towards environmental control in the form of electrons [8]: where the addition of chemical reagents is unnecessary, coagulation can be carried out by electrons. The mining industry requires a considerable amount of water for the processes of grinding and flotation, which is treated by chemical settling through the addition of reagents. This water is recirculated in a closed circuit. However, in each cycle, the water accumulates metallic ions and organic matter which are not removed, until the water reaches a saturation point, affecting the efficiency of the process in general and potentially leading to environmental issues when the tailing dam water accumulates.

Although electrocoagulation is not a new technology, it has not been developed beyond a pilot scale due to the long processing times that make the process more expensive. Despite this, electrocoagulation has become commercially important in the use of specific treatment of some pollutants, positioning itself as a technique with greater comparative advantages than those of traditional treatment technologies. Electrocoagulation is a process in which the suspended particles are destabilized by the induction of an electric current [9]. Additionally, a secondary reaction that can occur is the electrowinning of metals, decreasing the concentration of metallic ions present in the water [10]. The mechanism of electrocoagulation is highly dependent on the aqueous medium, especially its conductivity, and other characteristics such as pH and particle size [11]. Numerous studies have been carried out on electrocoagulation, but are often limited to the experimental investigation of the effect of these variables on the removal of pollutants [12]. In this work, electrocoagulation is proposed as an alternative for the treatment of mining unit process water to withdraw persistent ions that cannot be removed by traditional techniques. Despite a wealth of information on the subject, the mechanisms of electrocoagulation are not yet clear, and so in this study, a reaction mechanism is proposed and validated using a kinetic mathematic model.

## 2. Materials and Methods

The water analyzed in this study originated from a mining unit located in central Mexico and possessed an initial copper concentration of 150 ppm. The experiments were carried out in a cylindrical polypropylene reactor with a capacity of 10 L, using aluminum (Figure 1a) and stainless steel electrodes (Figure 1b) with a separation distance of 10 cm between them and 150 cm$^2$ total area. Aliquots were withdrawn at different times for the quantification of turbidity (NTU) and metal concentration. The pH was monitored through the electrochemical treatment with a pH meter equipped with a pH electrode Hanna. Copper was quantified using atomic absorption spectrophotometry (Buck Scientific V210). The turbidity was measured using an Orbeco TB200 turbidimeter (Orbeco-Hellige, Inc., Sarasota, FL, USA). For comparative analysis of the process water and electrochemically treated water's flotation efficiency, flotation tests were conducted using minerals from the same mine, with zinc, lead, copper, and silver intermetals and 2.39, 0.64, and 0.33% metallic content, respectively. The silver content was 56 g per ton. Aerofin and 1% isopropyl xanthate were used as collectors; zinc sulfate and sodium cyanide were used as depressants at 5% and 0.1%, respectively; and reactive teuton 240-like was used as a foaming agent. The process consisted of a primary flotation where lead concentrate and a tail were obtained,

and zinc concentrate was recovered. Prior to the primary flotation, two cleaning processes were performed to obtain the final concentrate.

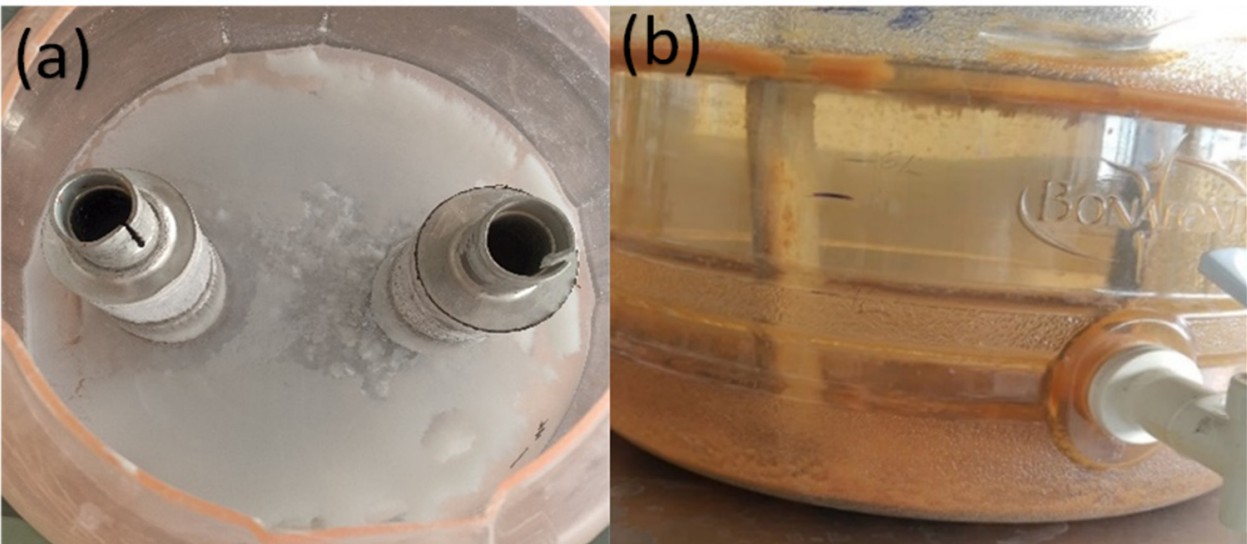

**Figure 1.** Electroreactor of polypropylene with 8 L volume for mine water electrocoagulation using two different electrodes: (**a**) aluminum electrodes and (**b**) stainless steel electrodes.

The flotation tests were conducted in an FLSmith rotation cell at 850 revolutions per minute. Microscope analysis was carried out using an electronic microscope SUPRA 40 made in Jena Germany for ZEISS equipped with graphite band, employing the program Quem-Scan to determine the species present in the clots.

## 3. Results

In the flotation tests of lead, zinc and copper, various reagents were used, such as copper sulfate, calcium hydroxide, and zinc sulfate, generating ions in the water which included Cu (II), Ca (II), and Zn (II). In flotation, the presence of high concentrations of copper ions reduces the efficiency of selective flotation. It is known that ingesting copper can cause serious illness, such as damage to the kidneys or liver [13], and therefore it is vital that these ions are removed from water. Water from mining processes is frequently treated using chemical settling, recovered, and recirculated; however, a drawback of this technique is that it does not eliminate the metallic ions present in the water, leading to an accumulation of metal ions which accumulate over each process cycle, modifying interfacial properties, and causing low efficiency in the flotation process. One technique used to remove metal ions is electroplating, which employs similar cells to those used in electrocoagulation; the main difference is that the anode undergoes dissolution in electrocoagulation, while this effect does not have a major impact on electroplating [14]. In both processes, the current density is crucial; it affects the amount of coagulant generated [15,16]. This variable can determine if the process is economically viable or not.

Figure 2 shows the percentage of residual copper in the water over time with different charge densities (104 A/m$^2$, 204 A/m$^2$, and 306 A/m$^2$). A reduction in the amount of copper in the water is shown for all densities. In this sense, according to Faraday's law, it is expected that at higher current densities, the speed of copper removal would be faster, which was true for the charge densities of 104 and 204 A/m$^2$, where the percentages of residual copper after electrolytic treatment were around 43% and 3%, respectively. However, in the case of the density of 306 A/m$^2$, a lower copper removal speed was found than that of 204 A/m$^2$, with a residual copper percentage of 31%. In this sense, Shokri and Karimi (2020) [17] found that, in electrochemical processes, an increase in current has a negative impact on the efficiency of pollutant removal.

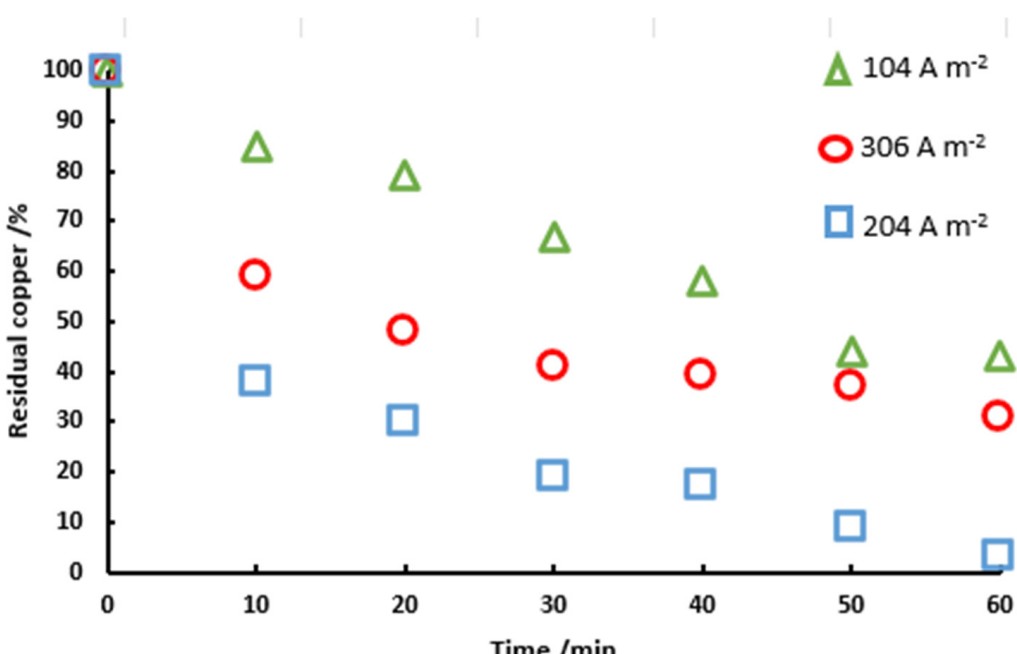

**Figure 2.** Residual copper content vs. time in 8 L of mining unit process water, using steel electrodes with different current densities (104, 204, and 306 A/m$^2$).

This may be due to the polarization of the electrode, since from 30 min onwards the percentage of residual copper remained relatively similar for the 306 A/m$^2$ charge density. It is known that polarization increases with the charge density [18].

Similarly, it is known that there is a critical current density value that, if exceeded, leads to electrical energy being wasted in water heating and the current efficiency decreasing [19].

On the other hand, mine process water contains organic particles added in the grinding and flotation processes, including necessary chemical reagents such as collectors, foaming agents, and conditioners, which cannot be completely removed using only the clarification process. However, the electrocoagulation process has been shown to be efficient for agglomerating suspended particles in water [7]. In this sense, turbidity is a measure of the clarity of a liquid, measuring the light that is trapped by the material suspended in the water. In turn, turbidity increases when matter agglomerates, increasing turbidity and generating larger particles. The scheme of this process is shown in Figure 3, beginning with the electrolytic dissolution of the anode (aluminum) (Equation (1)) to form cationic aluminum, in addition to the formation of the hydroxide ion at the cathode through the hydrolysis of water and in the presence of suspended particles, where the hydroxide ions and aluminum ions are attracted by charge forces between them (Yellow arrows in Figure 3) for forming aluminum hydroxide.

$$Al \rightarrow Al^{3+} + 3e^- \tag{1}$$

Figure 4 shows the stage at which the aluminum ions and hydroxides form aluminum hydroxide (Equation (2)), which destabilizes the suspended organic particles, causing them to suffer an attractive force from the surrounding aluminum hydroxide, thereby increasing the turbidity and interrupting the electric current.

$$Al^{3+} + 3OH^- \rightarrow Al(OH)_3 \tag{2}$$

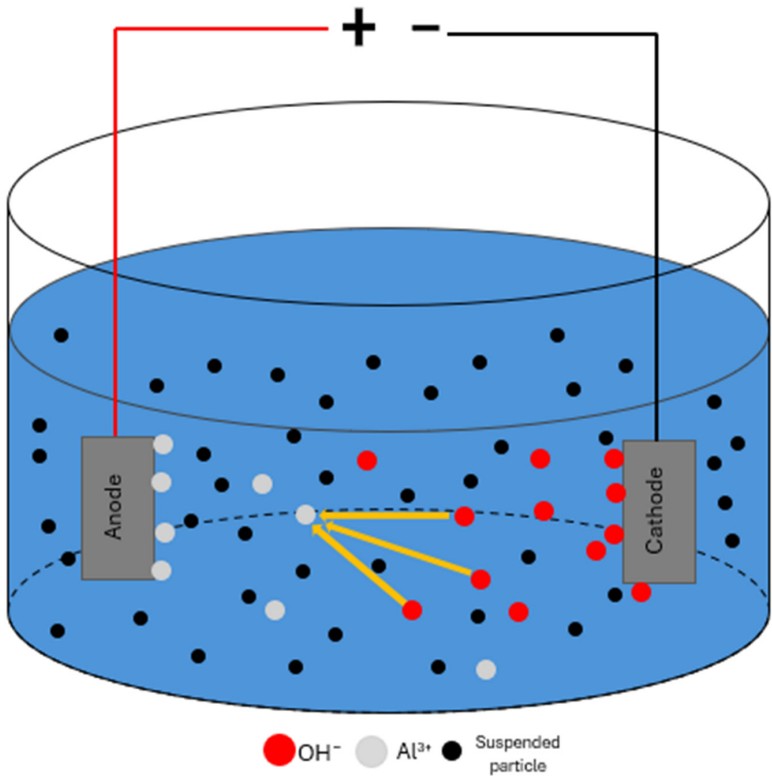

**Figure 3.** Initial electrocoagulation step: aluminum anode dilution and the formation of OH⁻ ions in the cathode.

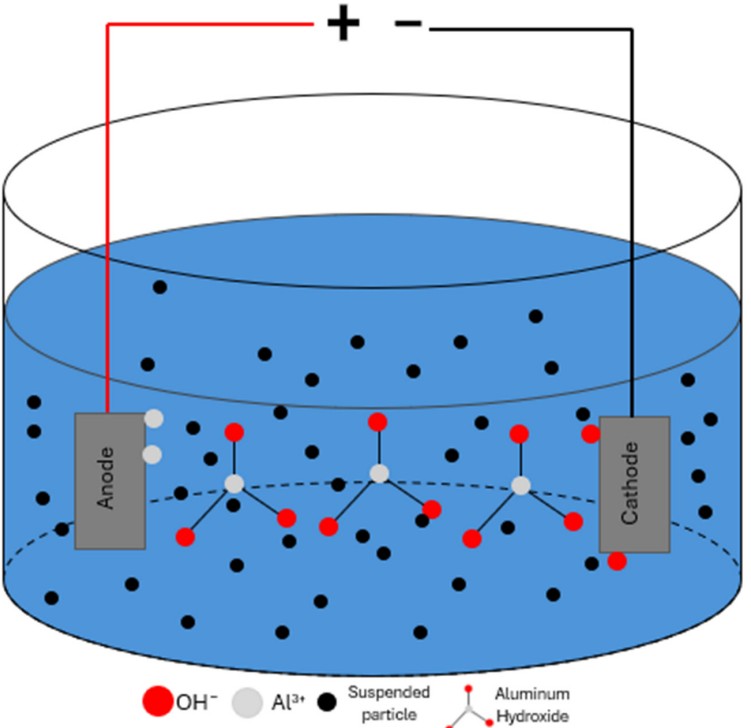

**Figure 4.** The hydroxide ions and aluminum form aluminum hydroxide.

After the current was interrupted, the formed clots began to settle at the bottom of the reactor, resulting in clarified water accumulating in the center of the reactor (blue part in Figure 5) and the slurry was materializing and settling in bottom of the cell (yellow part in

Figure 5), avoiding the need for a filtration stage. Sedimentation is considered to be the slowest stage of the process.

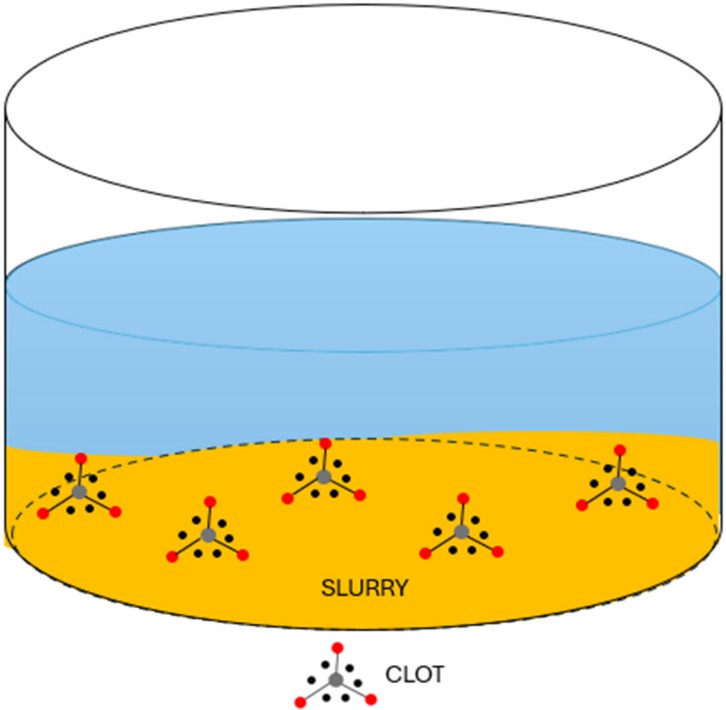

**Figure 5.** Formation of clots containing aluminum hydroxide to trap ions and organic matter which continued to settle.

Different authors have proposed empirical models, such as first-order, second-order, Langmuir, Elovich, and fractional models, to predict pollutant removal via electrocoagulation. However, these models were not mathematically developed based on the mechanism of the process [20]. In this sense, the chemical reaction of the electrocoagulation process is represented by the following equations:

$$Al \xrightarrow{k_1} Al^{3+} + 3e^- \tag{3}$$

$$3H_2O \xrightarrow{k_2} 3OH^- + 3H^+ \tag{4}$$

$$Al^{3+} + 3OH^- \xrightarrow{k_3} Al(OH)_3 \tag{5}$$

$$3H^+ + 3e^- \xrightarrow{k_4} \frac{3}{2}H_2 \tag{6}$$

$$Al(OH)_3 + \text{organic matter} \xrightarrow{k_5} NTU \tag{7}$$

Hess's law is applied to obtain the total reaction.

$$Al + 3H_2O + \text{organic matter} \xrightarrow{K} NTU + \frac{3}{2}H_2$$

According to this mechanism, a mathematic kinetic model is proposed where step (7) is as step controlling:

$$\frac{d[NTU]}{dt} = k_5[Al(OH)_3][\text{organic matter}]$$

Considering that $[Al(OH)_3]$ is a short-lived species, we used Equations (5) and (7) as follows:

$$\frac{d[Al(OH)_3]}{dt} = k_3\left[Al^{3+}\right]\left[OH^-\right]^3 - k_5[Al(OH)_3][organic\ matter]$$

$$\frac{d[Al(OH)_3]}{dt} = 0 = k_3\left[Al^{3+}\right]\left[OH^-\right]^3 - k_5[Al(OH)_3][organic\ matter]$$

$[Al(OH)_3]$ was solved as follows:

$$[Al(OH)_3] = \frac{k_3\left[Al^{3+}\right]\left[OH^-\right]^3}{k_5[organic-matter]}$$

and a similar method was used for $\left[Al^{3+}\right]$

$$\frac{d\left[Al^{3+}\right]}{dt} = k_1\left[Al^0\right] - k_3\left[Al^{3+}\right][OH]^3$$

$\left[Al^{3+}\right]$ was solved as follows:

$$\left[Al^{3+}\right] = \frac{k_1\left[Al^0\right]}{k_3[OH]^3}$$

By substituting $[Al(OH)_3]$ and $\left[Al^{3+}\right]$, the following was obtained:

$$[NTU] = k_1\left[Al^0\right]t$$

If

$$\left[Al^0\right] = 1;\ due\ Al\ is\ a\ solid\ and\ its\ activity\ is\ 1$$

since

$$k_1 = K$$

resulting in

$$[NTU] = Kt$$

A proportionality between the change in turbidity (NTU) and time was observed. The turbidity (NTU) of the water at different times and at different current densities is plotted in Figure 6, where an acceptable adjustment of the model to the experimental data can be seen. It can be seen that the process follows zero-order kinetics, and the current density has a more considerable effect.

As established above, the clarification of water consists of two stages: electrocoagulation and settling. Electrocoagulation is where colloidal particles come together to form clots, increasing their weight and sediment through gravity. Figure 7 shows the turbidity (NTU) vs. time after the electrochemical treatment, demonstrating a drastic decrease 60 min after suspending the electric current and reaching NTU values of 15.72, 15.57, and 14.79 for 129, 64.23, and 38.96 A/m$^2$, respectively, after 360 min of electrotreatment. This emphasizes that at lower current densities, due to the reduced current, fewer NTUs are formed, and the sedimentation of the clots is favored. In this sense, it has been found that there is an optimal current density in the water clarification process; once that value is reached, no improvement in solid removal efficiency is seen [19,21].

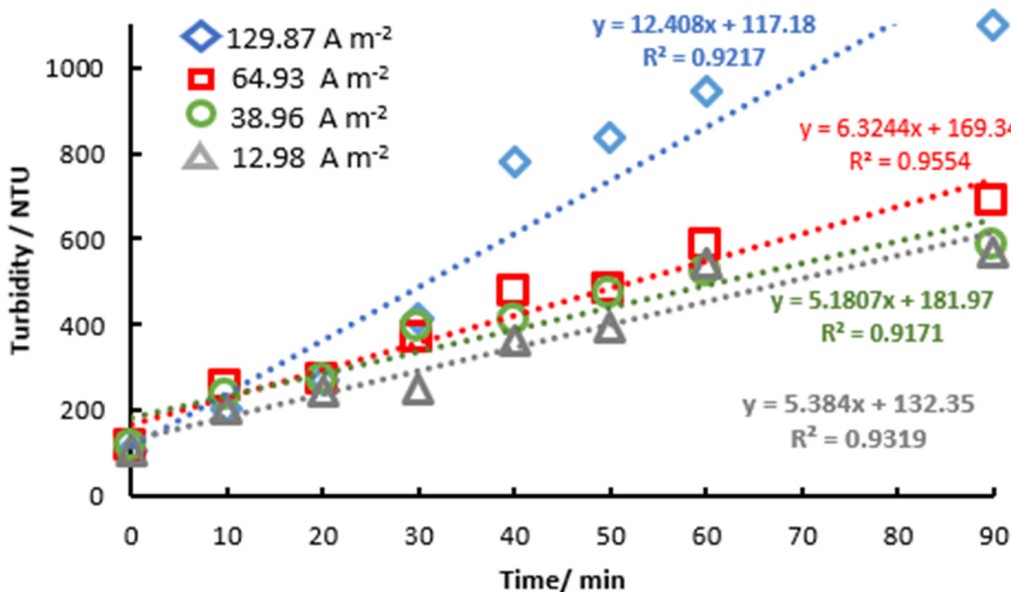

**Figure 6.** Turbidity over time throughout the electrocoagulation of mining unit water using aluminum electrodes and 8 L of water and the fit of the zero-order kinetic model to the turbidity of mining unit water.

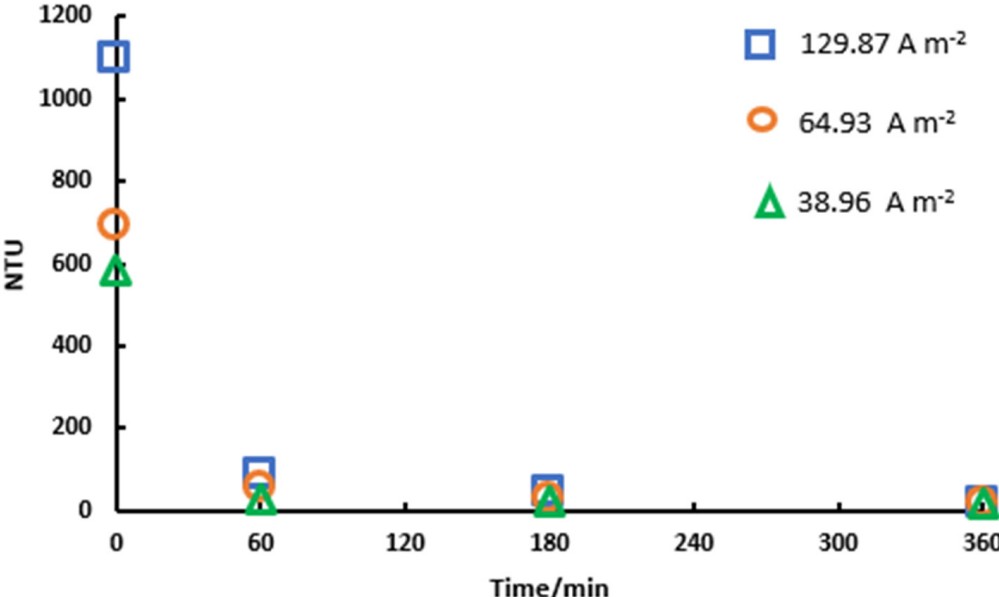

**Figure 7.** Amount of suspended solids (NTU) vs. time in 8 L mining unit water after suspension of the electric current using aluminum electrodes.

In this regard, Von Smoluchowki's model is frequently used to explain the kinetics of sedimentation through the actions of certain concentrations of electrolyte solutions (Equation (20)) [22]. In this model, it is proposed that the coagulating electrolytes compress the electrical double layer, which leads to a decrease in the repulsion forces between the colloidal particles, which unite upon collision to form binary and tertiary clots, and so on [12–24].

$$\frac{1}{v} = \frac{1}{v_0} + kt \tag{8}$$

where

$$v = \text{Clots concentration}$$

$$v_0 = \text{Clots initial concentration}$$

$$t = \text{time}$$

According to the proposed mechanism, coagulation and sedimentation tests were performed. Figure 8 shows the inverse of the turbidity vs. time in the electrocoagulation tests after cutting off the electric current. An acceptable fit of the model to the experimental data is shown, demonstrating a greater sedimentation kinetic constant and a lower current density. This is contrary to expectations, since according to Faraday's law, with a higher current density, a greater amount of aluminum hydroxide would be formed, which would favor coagulation. It was previously shown that electrocoagulation is not affected by the concentration of aluminum hydroxide that is formed since it follows zero-order kinetics, and the settling follows second-order kinetics.

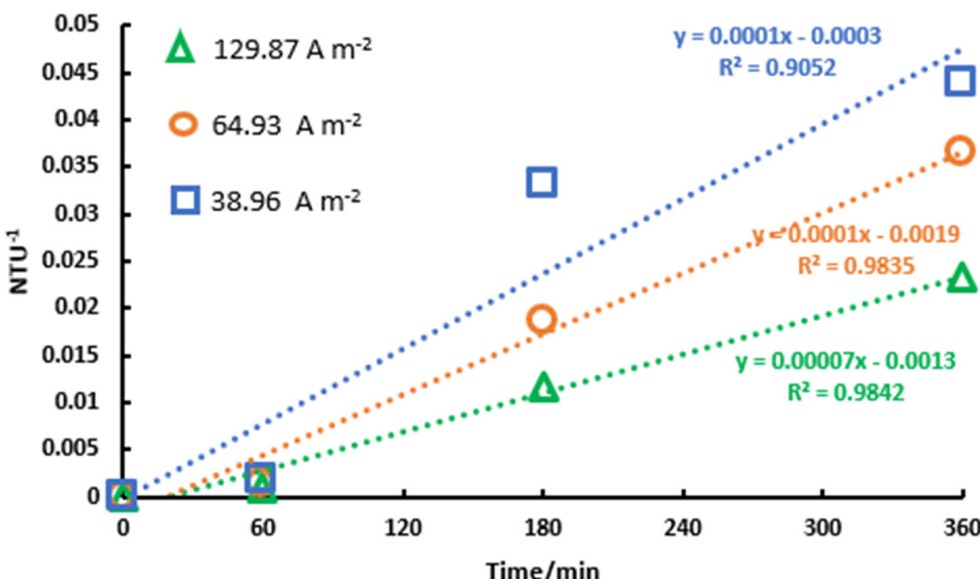

**Figure 8.** Fitting of Von Smoluchowki model: inverse of amount of NTU vs. time in 8 L of mining unit water using aluminum electrodes.

The clots formed during electrocoagulation were filtered and air-dried to be analyzed using Scanning Electron Microscopy (SEM). We found that the clots were composed of spongy matter, which corresponded mainly to aluminum and oxygen, with traces of silicon, calcium, and sulfur (Figure 9).

To a lesser extent, the presence of zinc, copper, lead, and arsenic minerals was detected using SEM-EDX.

A punctual analysis of the sample was also carried out to identify its size and chemical composition. The brightest particle corresponds to zinc, observing that it was trapped by the clot and had a particle size of 19.78 μm (Figure 10).

In addition, the presence of arsenic was observed in the clots, meaning that aluminum hydroxide can encapsulate toxic substances such as arsenic and be used for the efficient recovery of contaminated water. Also, it was found that, during electrocoagulation, crystals formed that contained Zn, Pb, Fe, Ca, and S (Figure 11), which are the main ions that cause flotation efficiency to decrease.

With our Scanning Electron Microscopy findings, the proposed mechanism of electrocoagulation (the formed clots encapsulate the ions present in the water, resulting in clarification) can be validated.

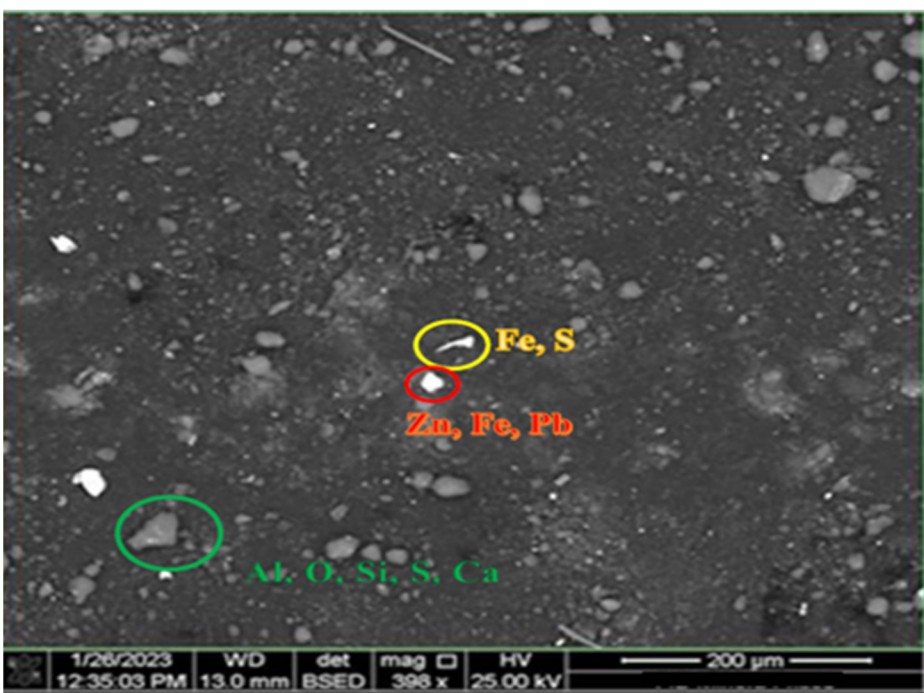

**Figure 9.** SEM photograph of the clots formed during the electrochemistry process in 8 L of mining unit water and aluminum electrodes magnified by 398×.

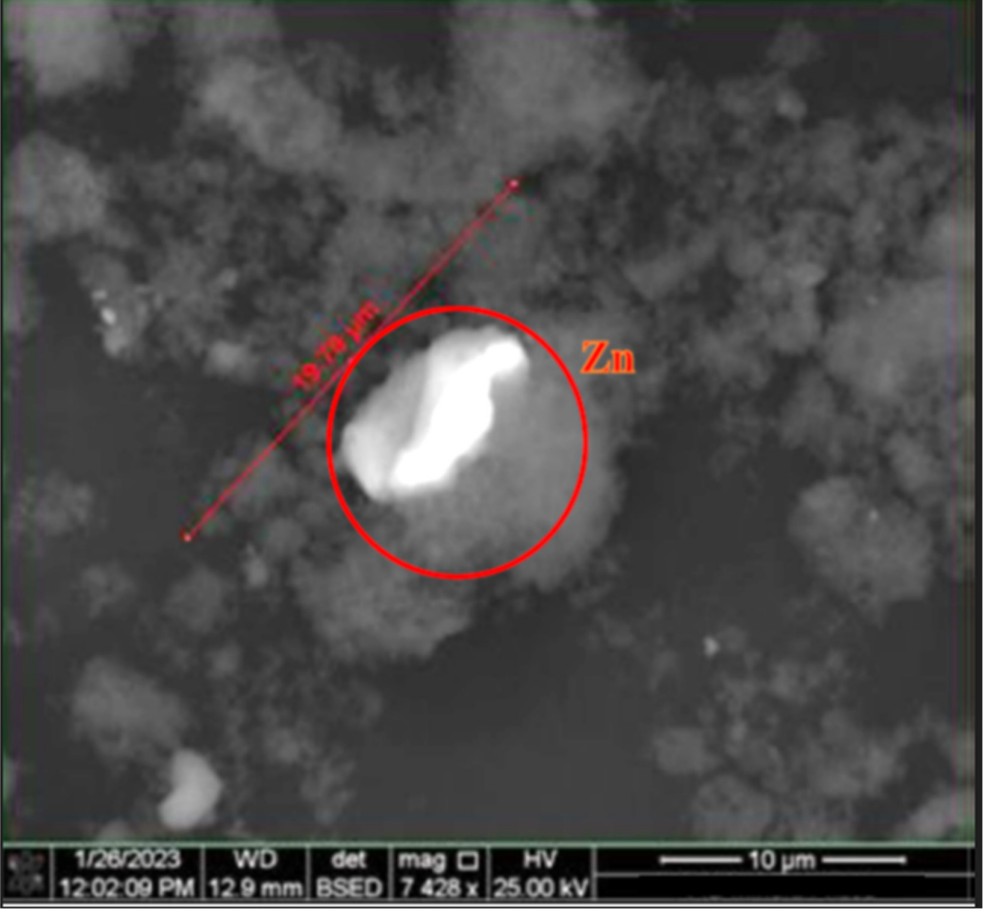

**Figure 10.** Punctual analysis of clots formed in electrocoagulation of 8 L of mining unit water, using aluminum electrodes, magnified at 7428×, resulting in the trapped zinc.

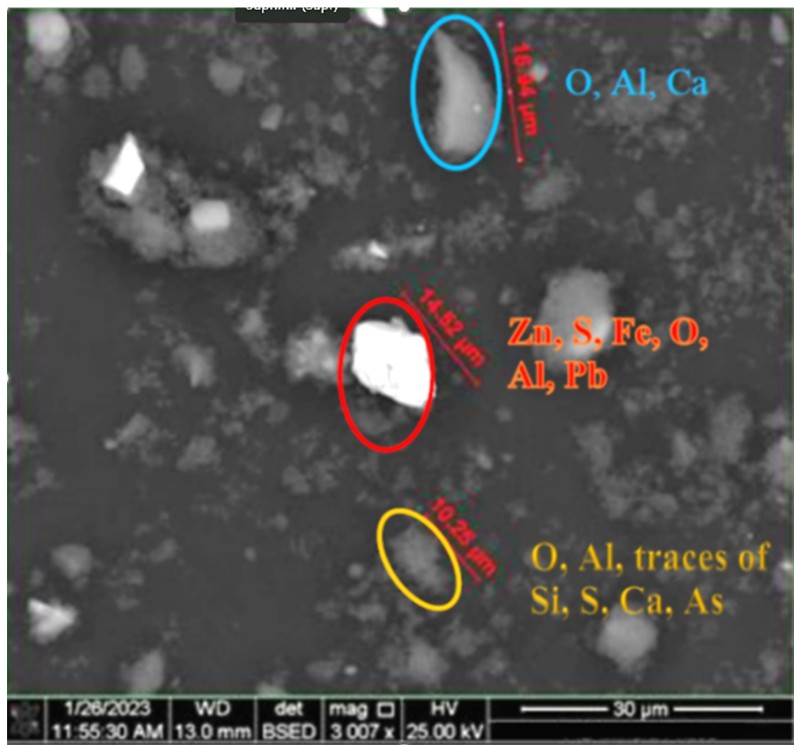

**Figure 11.** Punctual SEM analysis of clots formed in the electrocoagulation of 8 L mining unit water using aluminum electrodes, magnified at 3007×, resulting in the presence of Al, Ca, Zn, S, Pb, Sn, Si, and As.

*Flotation Test*

One of the principal goals of the electrochemical treatment of mining water is to increase the efficiency of metal recovery. Precious metals (Au and Ag) are of major interest and can be recovered via flotation. Flotation tests were carried out on both untreated process water and treated water. Figure 12 shows the recovery percentages of Zn, Pb, Au, and Ag in both water samples; it can be observed that, in the treated water, the recovery of Au and Ag increased by around 1.5%. This may not appear to be a large difference, but in industrial quantities of water, it is considerable, as the recovery of Zn from treated water was as high as 90% compared to only 60% recovery in untreated water, thereby establishing that electrocoagulation has a positive effect on metal recovery through flotation. In the case of lead, no considerable difference was observed.

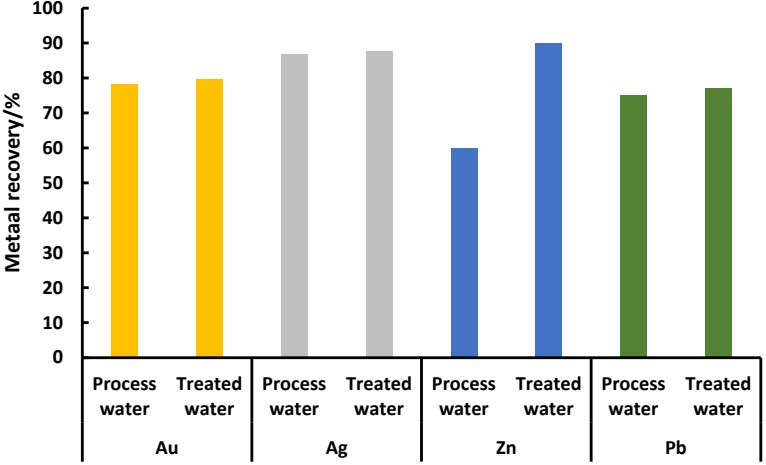

**Figure 12.** Recovery percentages in the flotation tests of Zn, Pb, Au, and Ag with the process and treated water, using aerofin and 1% isopropyl xanthate as collectors, zinc sulfate and sodium cyanide at 5% and 0.1%, respectively, as depressants, and reactive teuton 240-like as foaming agent.

## 4. Discussion

The application of electrocoagulation in the treatment of mining process water has been very poor due to its potential to remove persistent ions and molecules from water being relatively unknown. This method has been shown to be the most effective for removing heavy metals compared with other methods [6]. In this study, it was shown that electrocoagulation can achieve major clarification of mine process water. Most of the copper ions were removed with the use of iron electrodes. Achieving greater efficiency in metal flotation diminished the activation–depression effect of ions. The aluminum electrodes showed excellent results when capturing organic matter, and the SEM photograph shows that metallic ions were trapped in the clots formed. On the other hand, there is a lack of research on the mechanisms of electrocoagulation in the literature for formulating reactor models at the laboratory scale. However, there are a large number of studies involving simplified kinetic models that explain the difficulty of describing numerous physicochemical phenomena occurring simultaneously in electrocoagulation reactors. The models are based on important simplifications, such as the assumption of pseudo-equilibrium [12]. We propose a reaction mechanism, supported by a zero-order kinetic mathematic model, that demonstrates how electrocoagulation occurs and shows that electrocoagulation depends on time alone. Furthermore, one benefit of using electrocoagulation to treat water in the mining industry is the increased recovery of the precious metals Ag and Au (of an order of 1.5%), and a 30% increase in Zn froth flotation, which may indicate that ions and organic matter present in water inhibit the efficiency of the process.

The next step in this research work is to analyze the effect of the other variables in the electrocoagulation, such as pH, potential, and electrode distance, and then to propose the model to scale-up the experimental set-up to pilot plant size.

## 5. Conclusions

Electrocoagulation is a highly effective process for the treatment of mining water that can reduce the number of metallic ions present (e.g., Ca, Cu, and Fe), including toxic substances like arsenic. This can improve the efficiency of froth flotation, increasing the recovery percentage of the precious metals Ag and Au by 1.5% and the percentage of zinc recovery by 30%. A mechanism reaction was proposed that may explain electrocoagulation, consisting of five stages: (1) the dissolution of the anode, releasing metallic ions; (2) the formation of hydroxide ions in the cathode; (3) the formation of metallic hydroxide; (4) the trapping of organic matter in the clots; and (5) the settling of the clots. The first four steps involve the use of electric current for the formation of clots with metals and organic matter. The fifth is related to settling when the clot reaches the required size.

A kinetic mathematical model was formulated in order to validate the proposed reaction mechanism, and an acceptable fit with the experimental data was obtained. Zero-order kinetics were observed for the first stage, in which the clots are formed, and second-order kinetics were observed for the second stage, in which the settling occurs.

The SEM photograph of the formed clots agrees with the model, showing the formation of aluminum hydroxide and the trapping of metallic ions (Zn, Al, S, Pb, and Fe) and organic matter, as well as toxic substances like arsenic.

Electrocoagulation shows excellent applicability for clarifying mining process water, attaining turbidity levels of 14 NTU; the water in this experiment almost reached domestic quality.

**Author Contributions:** Conceptualization, O.J.S.-M. and A.T.-L.; methodology, O.J.S.-M. and R.Z.-G.; software, A.N.-B. and R.Z.-G.; validation, O.J.S.-M. and B.S.R.; formal analysis, A.T.-L. and J.P.R.-L.; investigation, J.A.H.-M. and J.P.R.-L.; resources, O.J.S.-M. and J.A.H.-M.; data curation, A.N.-B. and O.J.S.-M.; writing—original draft preparation, A.T.-L. and O.J.S.-M.; writing—review and editing, B.S.R. and O.J.S.-M. All authors have read and agreed to the published version of the manuscript.

**Funding:** The authors would like to thank the IPN Research Programs and Projects Administration System for the financial support No. 20231160 to carry out this work, and CONHACyT for the support provided to researchers through the National System of Researchers.

**Data Availability Statement:** Data are contained within the article.

**Acknowledgments:** The authors would like the IPN Research Programs and Projects Administration System for the financial support to carry out this work, and CONHCyT for the support provided to researchers through the National System of Researchers.

**Conflicts of Interest:** The authors declare no conflict of interest.

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
