# Peer review of "Clarification of Mining Process Water Using Electrocoagulation"

_minerals, doi:10.3390/min14040412_

Round 1

Reviewer 1 Report

Comments and Suggestions for Authors

Thank you for choosing to work on thus topic.The subject is highly in remote areas, water scarce areas in particular. Kindly see the following comments, hope that they will be of interest:

Line Number

 24 Aluminum is not considered environmental friendly, why not trying iron electrodes? Such a shift has been done in many municipal and industrial plants.

67 EC coagulation-flocculation mechanism is very similar to chemical coagulation mechanism + organics oxidation effect.

114 Kindly explain the reason for the sharp overdosing at 306 A/m2 which is uncommon at hydroxylic coagulants like alum.

144 Highly questionable, explain mechanism please. The large number of original particle of relatively  high specific area is being reduced to a smaller particles number with a smaller specific area, therefore it is expected to that turbidity would decrease. It should be also considered, that turbidity, being a course light scattering parameter, represents surface size and quantity of particles in the colloidal range, rather than large and flufy particles like aluminum hydraulic complexes.

 200 Not new, always exists in water treatment systems which include coagulation

 222 Could not it beeing expected in the beginnig due to the overdosing phenomena of coagulants.

304 It is kindly recommend to the respected authors to consider more  publications in scientific journals produced by water and wastewater researchers,  in order to enrich the article with EC of water and wastewater theory and application resulting from that world.

Comments on the Quality of English Language

I would encourage the respected researcher to improve on the grammar and scientific writing. 

Author Response

Thank you for choosing to work on thus topic.The subject is highly in remote areas, water scarce areas in particular. Kindly see the following comments, hope that they will be of interest:

Line Number

 “24 Aluminum is not considered environmental friendly, why not trying iron electrodes? Such a shift has been done in many municipal and industrial plants.”

Thank you for your observation, in fact, was carried out experiments with iron electrodes for withdraw of copper and for the organic Materia was used aluminum, as is described in thorough paper.  

“67 EC coagulation-flocculation mechanism is very similar to chemical coagulation mechanism + organics oxidation effect.”

Thank you for your comments. Yes we agree with this but in this row, that is analyzed is the effect of electrowon of copper and after the electrocoagulation.

“114 Kindly explain the reason for the sharp overdosing at 306 A/m2 which is uncommon at hydroxylic coagulants like alum.”

Thank you for your question. In this way, high values of current density are desirable in order to minimize the problems with side reactions and to provide a good energy. The current density should be the as high possible. Furthermore, the efficiency of elimination of one contaminant is increased when the current density is increased. [Walsh F. and Reade G., Design and Performance of Electrochemical Reactors for Efficient Synthesis and Environmental Treatment, Analyst, 1994, 119, 791-796. 

“144 Highly questionable, explain mechanism please. The large number of original particle of relatively  high specific area is being reduced to a smaller particles number with a smaller specific area, therefore it is expected to that turbidity would decrease. It should be also considered, that turbidity, being a course light scattering parameter, represents surface size and quantity of particles in the colloidal range, rather than large and flufy particles like aluminum hydraulic complexes.”

Thank you for your question. The mechanism is proposed that successes in two steps, in the first occurs the floc formation, which include 4 subsets, (1) dissolution of anode forming metallic ion, (2) formation of hydroxide ion in the cathode, (3) formation of metallic hydroxide, (4) after the metallic hydroxide formation the cots are agglomerated trapping the organic matter, producing intermedium size particle between microscopic and macroscopic size, which can’t see at simple vista, but the size is most large than a molecule, and secondary reaction occur of the molecular hydrogen formation. The other step successes when the current is off, and so the settling of flocs carried out.  

“ 200 Not new, always exists in water treatment systems which include coagulation”

Thank you for your comment. In the paper we mentioned in row 64 that the electrocoagulation isn´t a new process, but hasn´t a developed an industrial scale.

“ 222 Could not it beeing expected in the beginnig due to the overdosing phenomena of coagulants.”

Thank you for your observation. In this sense the literature stablishes a concentration work range for aluminum of the 1 to 4 g L-1, And the maximum aluminum concentration it has of the 0.125 g L-1, this calculate with Law Faraday and the volume reactor, it was under range stablished.

“304 It is kindly recommend to the respected authors to consider more  publications in scientific journals produced by water and wastewater researchers,  in order to enrich the article with EC of water and wastewater theory and application resulting from that world.”

Thank you for your recommendation. This were attended.

Reviewer 2 Report

Comments and Suggestions for Authors

In this paper, it is proposed that the electrocoagulation method is used for the purification of mineral processing wastewater, which can effectively remove metal ions and organic matter in mineral processing wastewater, and also has the removal effect on arsenic and other toxic elements, which can avoid the adverse effects of inevitable ions in wastewater on subsequent mineral processing experiments. On this basis, the flotation comparison test of Au and Ag was carried out with untreated and treated water, and the results showed that the flotation experiment of the treated wastewater by electrocoagulation method could increase the recovery rate of Au and Ag by 1.5% and 2%, which is of great industrial significance.

1.  In the preface of the article, the source and composition of the wastewater and the impact of using the wastewater to continue the beneficiation experiment are not described in detail.

2. In the materials and methods, it is mentioned that in order to compare the wastewater before and after treatment, the flotation experiment was used to separate lead and zinc, but the flotation experiment was not mentioned in the subsequent results, but the separation of Au and Ag was compared, which is questionable.

3.The effect of current density on the remaining amount of copper in wastewater is described in detail, but the article does not mention the specific adverse effects of copper or the specific separation of copper.

4. The advantages of flotation of wastewater treated by electrocoagulation should be described in detail, so as to help readers further understand the advantages of this method in treating beneficiation wastewater.

Author Response

  1. In the preface of the article, the source and composition of the wastewater and the impact of using the wastewater to continue the beneficiation experiment are not described in detail.

Thank you for your observation. This was added at the article in the 75 and 85 row.

  1. “In the materials and methods, it is mentioned that in order to compare the wastewater before and after treatment, the flotation experiment was used to separate lead and zinc, but the flotation experiment was not mentioned in the subsequent results, but the separation of Au and Ag was compared, which is questionable.”

Thank you for your recommendation. Information about the Zn and Pb test flotation is added in article 262-266 row.

“3.The effect of current density on the remaining amount of copper in wastewater is described in detail, but the article does not mention the specific adverse effects of copper or the specific separation of copper.”

Thank you for your comments.  The effect of copper is including in the article in 102 row.

  1. The advantages of flotation of wastewater treated by electrocoagulation should be described in detail, so as to help readers further understand the advantages of this method in treating beneficiation wastewater.

Thank you for your observation. And now the effect of treated the mine process water is shown in the article, in the 262 row.

Reviewer 3 Report

Comments and Suggestions for Authors

Dear Authors,

The submitted manuscript concern the very important issue of clarifing of  water in the very specific area of mining process. The topic of the article is very updated, however needs to be completed in terms of:

- Introduction: indication of the different applications of electrocoagulation in the removal of contaminants from water/ wastewater and also the effectiveness of this removing, especially in contects of mining process,

- Materials and Methods: a more detailed description of the physical and chemical properties and parameters of the analysed mining water,

- Discussion: please indicate and compare with other publications the use of electrocoagulation, the choice of parameters and its efficiency, especially in relation to other mine waters, as well as in general to waters/wastewater containing quality parameters similar to mine waters,

- Please indicate the strengths, weaknesses, concerns and risks of the electocoagulation process as well as the economic aspects of the implementation on an industrial scale the presented solution,

- Further prospects for research using this method and attempts to implement it in the near future.

Sincerely

Reviewer

Comments on the Quality of English Language

-

Author Response

“The submitted manuscript concern the very important issue of clarifing of water in the very specific area of mining process. The topic of the article is very updated, however needs to be completed in terms of: 
Thank you for the attention provided and the comments made.
“- Introduction: indication of the different applications of electrocoagulation in the removal of contaminants from water/ wastewater and also the efficiency of this removing, especially in contects of mining process,” 
Thank you for your comment, The paper points out the application of electrocoagulation in this technique. But it is proposed as an alternative for the removal of persistent ions, such as electrowinning. 
- Materials and Methods: a more detailed description of the physical and chemical properties and parameters of the analysed mining water, 
Thank you for your observations, now in the paper is included major information of the physical properties and parameters mining water.
- Discussion: please indicate and compare with other publications the use of electrocoagulation, the choice of parameters and its efficiency, especially in relation to other mine waters, as well as in general to waters/wastewater containing quality parameters similar to mine waters, 
Thank you for your recommendation, The paper included a discussion section, where the effectivity is discussed and the importance of the kinetic in this issue, only that the literature lacks 
studies of electrocoagulation with mining process water.
- Please indicate the strengths, weaknesses, concerns and risks of the electocoagulation process as well as the economic aspects of the implementation on an industrial scale the presented solution,
Thank you for your indication, in the discussion section is indicated the efficiency of the electrocoagulation in mining process water. Only that its implementation in the mining industry is scarce, being in an initial study stage. Which makes a precise economic study impossible. 
- Further prospects for research using this method and attempts to implement it in the near future.
Thank you for your recommendations, the prospects are found in the discussion section, where is shown that this work is the scale-up stage, for the implementation of this techinique.

Reviewer 4 Report

Comments and Suggestions for Authors

The manuscript was reviewed. The article is well-structured and written in appropriate English language and adds to knowledge and addresses to the readership of the journal. However, I suggest that the authors should address the following comments in the revised paper otherwise it must be rejected harshly:

1.      The introduction section should be rewrite.

2.      What is the difference between the study and similar studies, it is not adequately explained. Thus, a comprehensive discussion regarding the novelty of present work compared to other ones and knowledge gaps is suggested.

3.      It is suggested that the quality and resolution of all Figs. enhanced.

4.      The major shortcoming of the paper is that for optimization of critical factors interaction effects of each parameter was not explored. OFAT (one factor at time) method is not effective way for parameter optimization as all the other factors are kept constant. However, deign of experiments (DOE) is an efficient tool for optimization. The reason for using OFAT instead of DOE should be explored. The following references can be used and cited in the revised manuscript.

Investigation of UV/H2O2 process for removal of ortho-toluidine from industrial wastewater by response surface methodology based on the central composite design

Treatment of Aqueous Solution Containing Acid red 14 using an Electro Peroxone Process and a Box-Behnken Experimental Design.

https://doi.org/10.1016/j.surfin.2020.100705

·  https://doi.org/10.1080/03067319.2020.1791328

·  https://doi.org/10.1016/j.psep.2023.02.077

5.      the English status should be improved.

6.      the repeated sentences should be removed.

7.      It may be appropriate to review all units (such as mg/L or mgL-1) as standard.

8.      Captions of the figure and tables must be with complete information, conditions etc.

9.      Please follow the journal's guidelines for the format of materials and method.

10.  Please improve the quality of all figures (font too small and unreadable).

11.  All the equations should be checked once again, they should be both charge and mass balanced, please replace 1 ½   with 3/2, the mentioned format is incorrect..

12.  Please add the practical applications and future research prospects of this work before the conclusions section.

13.  A discussion of challenges that this study overcome and also potential remaining challenges must be conducted in a separate section. Thus following articles are related to your work and can be further analyzed and cited in the manuscript:

https://doi.org/10.1016/j.chemosphere.2022.133817

https://doi.org/10.1080/03067319.2020.1791328

The treatment of spent caustic in the wastewater of olefin units by ozonation followed by electrocoagulation process

Removal of ortho-toluidine from industrial wastewater by UV/TiO2 process

https://doi.org/10.1016/j.jiec.2024.01.058

https://doi.org/10.1007/s13762-023-05149-4                      

14.  Abstract section needs significant modifications in terms of enriching major findings of paper.

15.   What is the hypothesis of the study? Please clarify in the introduction. Also, the objective and novelty of the work should be clarified clearly.

Comments on the Quality of English Language

the English language should be improved.

Author Response

“The manuscript was reviewed”.

Thank you very much four kind attention to our material.

“The article is well-structured and written in appropriate English language and adds to knowledge and addresses to the readership of the journal.”

Thnak you very much, we appreciate your encouraging comments.

 “However, I suggest that the authors should address the following comments in the revised paper otherwise it must be rejected harshly”:

Thank you very much for your comments, we will attend every one of your comments carefully.

  1. The introduction section should be rewrite.

We appreciate your kind comments. We have attended your indication, and the now the introduction section was structured in the following form: problem statement, brief discussion about the achievements of the others research groups and finally the goal of the papers. In this way, we have attended the recommendation of the reviewer.

  1. “What is the difference between the study and similar studies, it is not adequately explained.”

Thank you very much for your important question. Certainly, our paper is novel due it deals about the treatment of process wastewater of mine by electrocoagulation which has not been studied previously by other authors. Then, our paper is very important contribution to technical and scientific knowledge, because it contains a proposal of kinetic mechanism of electrocoagulation.

  1. “Thus, a comprehensive discussion regarding the novelty of present work compared to other ones and knowledge gaps is suggested.”

Thank you for your comments. Certainly, our paper is novel due it deals about the treatment of process wastewater of mine by electrocoagulation which has not been studied previously by other authors. Then, our paper is a very important contribution to technical and scientific knowledge, because it contains a proposal of kinetic mechanism of electrocoagulation.

  1. “It is suggested that the quality and resolution of all Figs. enhanced.”

Thank you for your indication. We have carefully improved the quality in resolution of all figures as you can see in the revised version of the paper.

  1. “The major shortcoming of the paper is that for optimization of critical factors interaction effects of each parameter was not explored. OFAT (one factor at time) method is not effective way for parameter optimization as all the other factors are kept constant. However, deign of experiments (DOE) is an efficient tool for optimization. The reason for using OFAT instead of DOE should be explored. The following references can be used and cited in the revised manuscript.”

Thanks for your question. It is important to bring to the attention to reviewer that in this paper the goal is not to do optimization of the parameters of the processes. In this step of our research work, our objective is to investigate only the effect of the current density. In the next step we will analyzes the effect of the other variables and then do optimized design of the experiments. We thank your suggestion of new references which have including in the revised version paper.

“Investigation of UV/H2O2 process for removal of ortho-toluidine from industrial wastewater by response surface methodology based on the central composite design.”

“Treatment of Aqueous Solution Containing Acid red 14 using an Electro Peroxone Process and a Box-Behnken Experimental Design.”

https://doi.org/10.1016/j.surfin.2020.100705

  • https://doi.org/10.1016/j.psep.2023.02.077
  1. “the English status should be improved.”

We thank your indication and we have hired an expert in English and now the paper has been improved.

  1. “the repeated sentences should be removed.”

Thank you for your observation and we have removed the repeat sentences in the revised version of the paper.

  1. “ It may be appropriate to review all units (such as mg/L or mgL-1) as standard.”

Thank you for your observation, all units are standard.

  1. “Captions of the figure and tables must be with complete information, conditions etc.”

Thank you for your observation. The captions of the figures and tables contain the complete information.

“9.      Please follow the journal's guidelines for the format of materials and method.”

Thank you for ‘your comments. The format now is in the journal´s format.

“10.  Please improve the quality of all figures (font too small and unreadable).”

Thank you for your observation. The figures were improved, like you can see in the paper.

“11.  All the equations should be checked once again, they should be both charge and mass balanced, please replace 1 ½   with 3/2, the mentioned format is incorrect..”

Thank you for your observation. The equations were revised and corrected in the correspondence case.

  1. “Please add the practical applications and future research prospects of this work before the conclusions section.”

Thank you for your observation. And the next step in this research work is to analyze the effect of the other variables in the electrocoagulation such as, pH, potential, electrode distance, and then to propose the model to scale-up the experimental set-up to pilot plant size.

  1. “A discussion of challenges that this study overcome and also potential remaining challenges must be conducted in a separate section.”

Thank you very much for your suggestion. Certainly, the challenge that this study overcome are: 1) withdraw ions (Cu, As, Fe) presents in the water process mine, persistent organic molecules 2) The lack of electrocoagulation mechanism. 3) The lack of kinetic model to formulate a reactor model at laboratory scale. This discussion is present now in the revised version of the paper.     

Thus following articles are related to your work and can be further analyzed and cited in the manuscript:

https://doi.org/10.1016/j.chemosphere.2022.133817

https://doi.org/10.1080/03067319.2020.1791328

The treatment of spent caustic in the wastewater of olefin units by ozonation followed by electrocoagulation process

Removal of ortho-toluidine from industrial wastewater by UV/TiO2 process

https://doi.org/10.1016/j.jiec.2024.01.058

https://doi.org/10.1007/s13762-023-05149-4

Thank you for your recommendation, the articles had been revised and added in the paper.        

“14.  Abstract section needs significant modifications in terms of enriching major findings of paper.”

Thank you for your recommendation. The findings were added to article in abstract section.

  1. “What is the hypothesis of the study? Please clarify in the introduction. Also, the objective and novelty of the work should be clarified clearly.”

Thank you four your recommendation. The hypothesis and objective were added in the introduction in row 78.  

Round 2

Reviewer 3 Report

Comments and Suggestions for Authors

The content of the article has been revised very thoroughly and meticulously.

Reviewer 4 Report

Comments and Suggestions for Authors

The revised manuscript was upgraded and it can be accepted for publication in its current form 

Comments on the Quality of English Language

Mm